# How growers make decisions impacts plant disease control

**Rachel E. Murray-Watson**[1]*, **Frédéric M. Hamelin**[2], **Nik J. Cunniffe**[1]

**1** Department of Plant Sciences, University of Cambridge, Cambridge, United Kingdom, **2** IGEPP, INRAE, Institut Agro, Univ Rennes, Rennes, France

* rm844@cam.ac.uk

## Abstract

While the spread of plant disease depends strongly on biological factors driving transmission, it also has a human dimension. Disease control depends on decisions made by individual growers, who are in turn influenced by a broad range of factors. Despite this, human behaviour has rarely been included in plant epidemic models. Considering Cassava Brown Streak Disease, we model how the perceived increase in profit due to disease management influences participation in clean seed systems (CSS). Our models are rooted in game theory, with growers making strategic decisions based on the expected profitability of different control strategies. We find that both the information used by growers to assess profitability and the perception of economic and epidemiological parameters influence long-term participation in the CSS. Over-estimation of infection risk leads to lower participation in the CSS, as growers perceive that paying for the CSS will be futile. Additionally, even though good disease management can be achieved through the implementation of CSS, and a scenario where all controllers use the CSS is achievable when growers base their decision on the average of their entire strategy, CBSD is rarely eliminated from the system. These results are robust to stochastic and spatial effects. Our work highlights the importance of including human behaviour in plant disease models, but also the significance of how that behaviour is included.

**Data Availability Statement:** Sample code, along with the data used to make each figure, is available on: https://github.com/RachelMurray-Watson/How-growers-make-decisions-impacts-plant-disease-control..git

## Author summary

Models of plant disease epidemics rarely account for the behaviour of growers undertaking management decisions. However, such behaviour is likely to have a large impact on disease spread. Growers may choose to participate in a control scheme based on the perceived economic advantages, acting to maximise their own profit. Yet if many growers participate in a control scheme, their participation will lower the probability of others becoming infected and consequently disincentivise them from participating themselves. How these dynamics play out will alter the course of the epidemic. We incorporate these economic considerations into an epidemic model of Cassava Brown Streak Disease using two broad approaches, which vary in the amount of information provided to growers. We also consider the effect of grower misperception of economic and epidemiological parameters. Our work shows that both the inclusion of grower behaviour, and its means of

**Funding:** REMW acknowledges the Biotechnology and Biological Sciences Research Council of the United Kingdom (BBSRC; https://bbsrc.ukri.org/) for support via a University of Cambridge DTP PhD studentship (Project Reference 2119272). The funders had no role in study design, data collection and analysis, decision to publish, or preparation of the manuscript.

**Competing interests:** The authors have declared that no competing interests exist.

inclusion, affect disease dynamics, and highlights the importance of including grower decision-making in plant epidemic models.

## 1 Introduction

Human behaviour is intuitively important in the control of plant infectious disease, as individuals often face choices when adopting infection-limiting behaviours. This behaviour has been widely considered in the context of human (e.g. [1–5]; reviewed in [6] and [7]) and animal (e.g. [8–11]) diseases, though there have been fewer studies for plant diseases (with the exception of [12–16] and [17]). Where studies do consider individual's decisions there is much variation in how decision-making is modelled, although it clearly will affect the outcomes of their decisions.

We assume that decisions regarding disease control are intrinsically linked to disease prevalence. Clearly, they must also depend upon considerations such as the cost of control, the consequences of infection and the perceived risk of becoming infected [18–20]. Additionally, and crucially, decisions made by one grower will influence the decisions of another. Voluntary disease control programmes can therefore be viewed as a collective action problem (also termed social dilemmas; [21]) as they generate externalities that affect the outcomes of those not involved in the programme. By reducing or eliminating their own chance of infection, growers who control also lower the probability that their neighbours will be infected, thereby generating a positive externality that disincentives their neighbours' engagement in disease prevention [22].

Such "free-riding" behaviour is indirectly encouraged when there is higher overall participation in control schemes, as growers are less likely to become infected whenever the proportion controlling is high. Yet if too many growers engage in free-riding behaviour, there will be a resurgence of infections, which may make disease control more likely. These feedback loops are believed to be a major reason why some public health schemes for human disease, such as subsidised vaccination campaigns, have failed to lead to disease eradication [22]. Consequently, the success of voluntary control policies may be self-limiting. Indeed, perceived risk of other growers benefiting from an individual's control policies are a prominent reason why growers do not participate in control schemes [18, 23].

The strategic nature of grower behaviour can be modelled using game-theoretic analysis. Game theory is used to study decision making in settings where an individual (or "player") can choose from a variety of strategies, with the aim of choosing one that will maximise their desired outcome ("payoff") [24]. No individual makes a decision in isolation, and the best strategy to play will depend on the strategy choices of others. In our work, we assume growers have access to different combinations of three pieces of information: their own yield from the previous year, an estimate of the expected profits of each strategy (control or non-control) and the overall expected profit of the population. They then compare a subset of these different quantities to assess profitability and thus whether they should change strategy. Though the assumption that growers have access to "perfect information" is limiting and unlikely to bear true in a real-world context, understanding the results in such a case is still a helpful tool in studying the effects of these assessments on participation in control. The case of "perfect information" also acts as a useful baseline from which other scenarios where growers have less information may be assessed.

Game theoretic approaches have been used in previous disease models, often based on evolutionary game theory (e.g. [25] and [10]; for non-epidemic uses see: [26–31]). In this

formalism, in a two-strategy model, the performance of the player's own strategy is compared against that of the average performance of the population. If the payoff is less than that of the population, the focal player changes strategy; if it is higher, they remain with their current strategy. For plant disease, this would involve growers comparing the average profit of all of those using their own control strategy with the average profit over all growers in the population (we call this the "strategy vs. population" model). Of course this assumes, perhaps unrealistically, that growers will base their decision making upon the profit of all those using the same control strategy.

An alternative means of assessing profitability, is used in [12, 13] and [17] (though the inclusion of grower characteristics in [13] means it deviates from pure game-theoretic assumptions of rationality as growers may adopt a strategy that will earn them lower profits). In these two-strategy models, growers compare their own profit with the average profit of those playing the alternative strategy (the strategy that the grower does not currently use; for a controller, this would be non-control and vice versa). We henceforth call this the "grower vs. alternative" model. Ostensibly this seems a more likely comparison a grower will make, as it requires less information and focuses more closely on a grower's own performance, rather than all who act similarly to that grower. Other possible models combine elements of both approaches, since growers compare their strategy with the expected profit of the alternative strategy ("strategy vs. alternative" model) or compare their own profit with the average of the population ("grower vs. population" model). To our knowledge, the "grower vs. population" and "grower vs. alternative" models have never been compared.

The two model classes ("strategy vs." and "grower vs.") can be broadly classified based on the economic concepts of "rational" and "adaptive" expectations, respectively [32]. If an individual forms their expectations rationally, they assess the information available at any instant and then extrapolate/determine what that means for the future. This has a clear relationship with our "strategy vs." models, which use the current probabilities of infection as a proxy for what the future probabilities will be. Conversely, adaptive expectations are based solely on historical outcomes. Rather than simply being "adaptive", our "grower vs." models can instead be viewed as "strategic-adaptive" [33], accounting for prior experience whilst still accounting for future events.

Hypothetical scenarios in which each of these model types could potentially apply can be constructed. If a grower is part of an area-wide control scheme (such as the citrus health management areas (CHMAs) for citrus diseases in the United States of America; [34], or to manage Queensland fruitfly populations in Australia, [35]) that only includes a subset of the population, they may wish to compare the profits of the scheme with the expected profit of the entire population (the "strategy vs. population" comparison). The "strategy vs. alternative" comparison may be used if a grower is part of an area-wide control scheme but wishes to join another scheme with a different means of control. If a particular strategy may confer a market advantage to the grower (for example, if pesticide use is widespread, they may wish to stop spraying to then market themselves as "organic"), the grower may compare their outcome with the expected profit of the population ("grower vs. population"). Finally, if a grower is using a particular control strategy and then approached by an extension worker who proposes an different strategy, they may employ the "grower vs. alternative" comparison.

To examine the effects of these different models of growers' decision making in a concrete setting, we use a simple model describing the spread of Cassava Brown Streak Disease (CBSD) based on the model presented in [13]. Cassava (*Manihot esculenta*) is a staple food in Sub-Saharan Africa, where it is predominantly grown as a subsistence crop. Cassava production is threatened by CBSD, a viral disease caused by either cassava brown streak virus (CBSV, [36]) or Ugandan CBSV (UCBSV; [37]). Infection results in necrosis of the stems and tubers, leading

to yield losses of up to 70% [38]. Coupled with its increasing spread from east to west Africa, this positions CBSD as a major challenge to cassava cultivation across the region [39].

Both viruses are transmitted horizontally by a whitefly vector (*Bemisia tabaci*, [40]) and vertically through the replanting of infected stem cuttings. Informal trade of cassava cuttings is widespread amongst growers, though the subtlety of symptoms often means that the practice often contributes to vegetative propagation of infected material [41] and spread of CBSD between growers [42].

These potential for vertical transmission means that clean seed systems (CSSs) are a plausible CBSD management strategy [43]. CSSs disseminate virus-free planting material, thereby avoiding the vegetative propagation of infected material. The success of CSSs will depend on grower behaviour, as participation will be governed by the balance of costs and benefits of the scheme. This, and the combination of trade- and whitefly-mediated pathogen transmission, means that a grower's profit will depend on the action of others.

The overall objective of this work is to investigate how including grower behaviour in plant epidemic models affects the outcome of control schemes, and how the factors influencing behaviour can best be manipulated to encourage control uptake. We use our behavioural model alongside the CBSD case study to address the following: (1) How does profit comparison's formulation (i.e. "strategy vs." or "grower vs.") affect the model and its results, and can a scenario where all growers use the CSS be attained? (2) How does grower participation in the CSS depend on epidemiologically and economically important parameters, and how can high levels of control be encouraged? (3) How does systemic uncertainty in epidemiological parameters affect participation in the CSS? Though the "strategy vs." models are mathematically tractable and therefore allow for the derivation of analytical expressions, our view is that it is less likely that growers will make their assessment of profitability based on every other grower using the same strategy as they are, rather than just their own outcomes. Thus, after our initial comparisons (Question 1 above), we focus solely upon the "grower vs. alternative" model to investigate the effect of parameters and systematic uncertainty on CSS participation (Questions 2 and 3 above).

## 2 Materials and methods

### 2.1 Epidemiological model

Our model (Fig 1A) is a simplified representation of a CSS. Fields are classified by infection status (susceptible, $S$, or infected, $I$), as well as whether their growers currently control, i.e. used certified clean seed the last time the field was planted (subscripts: controller, $C$, or non-controller, $N$). We assume each grower cultivates only a single field, and so use the terms "grower" and "field" synonymously. Infection status and control status are binary classifications; we do not model within-field spread of disease, and assume controllers plant only certified virus-free material.

Susceptible fields are vulnerable to horizontal infection (at rate $\beta$) due to viruliferous vectors moving from infected fields (fields of class $I_C$ and $I_N$) (Table 1). Vertical infection also occurs via infected propagation material. We assume that the propagation material used by controllers is never infected. The probability of a non-controller becoming vertically infected depends jointly on the probability of acquiring infected planting material from an individual infected field ($p$) and the prevalence of disease at the time of planting $\left(\frac{I_C+I_N}{N}\right)$. The season length ($1/\gamma$) is independent of the control strategy adopted in, and infection status of, any given field. Since we use a continuous time model, fields are asynchronously harvested and replanted [44], a plausible assumption for cassava cultivation in much of Sub-Saharan Africa [45].

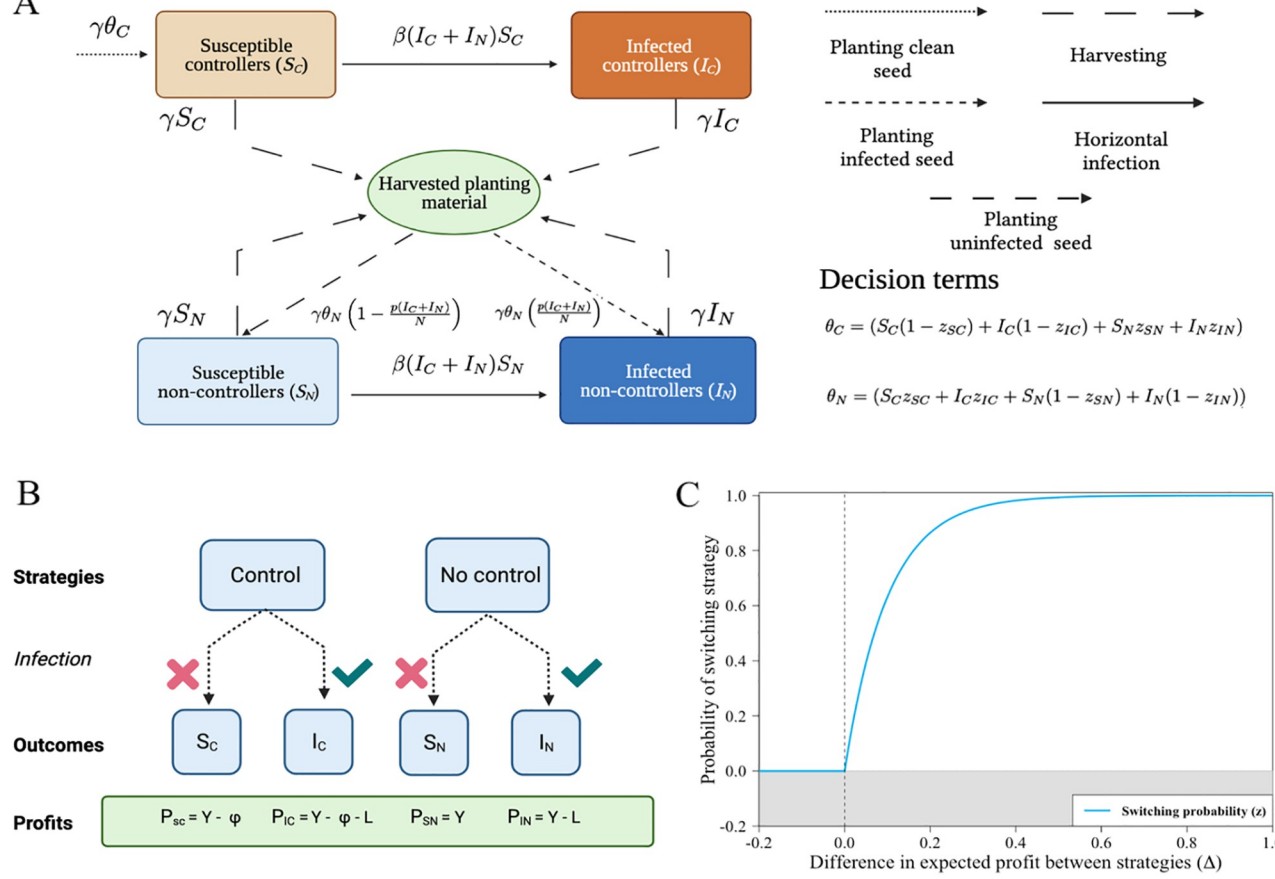

**Fig 1. Epidemiological and behavioural models.** A The epidemiological model distinguishes growers who control (i.e. plant clean seed) from those who do not. Only non-controllers can become infected via vertical transmission, although all growers are subject to horizontal transmission due to movement of vectors from infected fields (Eqs 5–8). When replanting growers potentially switch strategies. B Strategies, outcomes and profits; the latter account for the loss of yield if infected and the cost of participating in the control scheme. C Decisions to switch are based on a switching function, parameterising the probability of switching strategy $z$ as a function of $\Delta$ as difference in predicted profits. Fig A-B were created with BioRender.com.

**Table 1. Summary of parameter values.**

| Parameter | Meaning | Value | Reference |
|---|---|---|---|
| $1/\gamma$ | Length of the growing season | 300 days | [13]; [46] |
| $\beta$ | Rate of secondary infection | $8.93 \times 10^{-6}$ day$^{-1}$ field$^{-1}$ | Calibrated to [13] |
| $p$ | Probability of planting infected cuttings | 0.8 | Assumed (see main text) |
| $\eta$ | Responsiveness of growers | 10 | Assumed (see main text) |
| $Y$ | Maximum yield | 1 | All values scaled relative to yield |
| $L$ | Loss due to infection | 0.6 | [38]; [47] |
| $\phi$ | Cost of control | 0.25 | [13] |
| $N$ | Total number of fields/growers | 750 | Illustrative |
| $S_C(0)$ | Initial proportion of susceptible controllers | $0.1N$ | Illustrative |
| $I_C(0)$ | Initial proportion of infected controllers | 0 | Illustrative |
| $S_N(0)$ | Initial proportion of susceptible non-controllers | $0.89N$ | Illustrative |
| $I_N(0)$ | Initial proportion of infected non-controllers | $0.01N$ | Illustrative |

To integrate behavioural dynamics, we represent the decisions made by growers at the time of planting by *switching terms*. These reflect the probability of growers switching strategy for their following crop, based on their current strategy and the current state of the system, as well as, potentially, whether their previous crop was infected. In general, we distinguish four switching terms (Fig 1A).

$$z_{SC} = \mathbb{P}(\text{switch from } C \text{ to } N | \text{previous crop remained uninfected } (S)), \tag{1}$$

$$z_{IC} = \mathbb{P}(\text{switch from } C \text{ to } N | \text{previous crop was infected } (I)), \tag{2}$$

$$z_{SN} = \mathbb{P}(\text{switch from } N \text{ to } C | \text{previous crop remained uninfected } (S)), \tag{3}$$

$$z_{IN} = \mathbb{P}(\text{switch from } N \text{ to } C | \text{previous crop was infected } (I)). \tag{4}$$

As described below, the probabilities encoded in the switching terms depend on growers' assessments of likely profits, and thus depend on the comparisons we assume are made by growers.

Our general model is

$$\frac{dS_C}{dt} = \gamma\theta_C - \beta S_C(I_C + I_N) - \gamma S_C, \tag{5}$$

$$\frac{dI_C}{dt} = \beta S_C(I_C + I_N) - \gamma I_C, \tag{6}$$

$$\frac{dS_N}{dt} = \gamma\theta_N\left(1 - \frac{p(I_C + I_N)}{N}\right) - \beta S_N(I_C + I_N) - \gamma S_N, \tag{7}$$

$$\frac{dI_N}{dt} = \gamma\theta_N\left(\frac{p(I_C + I_N)}{N}\right) + \beta S_N(I_C + I_N) - \gamma I_N. \tag{8}$$

where:

$$\theta_C = S_C(1 - z_{SC}) + I_C(1 - z_{IC}) + S_N z_{SN} + I_N z_{IN} \tag{9}$$

$$\theta_N = S_C z_{SC} + I_C z_{IC} + S_N(1 - z_{SN}) + I_N(1 - z_{IN}) \tag{10}$$

## 2.2 Parameterisation

Baseline parameter values for CBSD were taken from previous studies where possible (Table 1). The rate of secondary infection, $\beta$, was set by matching the results of our model against [13]. In particular, we used the baseline behaviour of that model to motivate the choice that, in the absence of trade-mediated transmission, with 1% of fields initially infected and with no option of controlling for disease, 50% of fields would be infected within 10 seasons. To only account for the horizontal transmission, the probability of vertical transmission, ($p$) was set to zero when matching these results.

The value of 60% of fields' potential profit lost due to infection was assumed, to account for both the loss of starch content—which causes up to 40% loss of yield [47]—and the reduced market value of infected materials [38].

Vertical transmission requires cuttings taken from infected plants to be replanted and cause infection in the next season. The probability of vertical transmission is therefore reduced by selection, in which growers avoid replanting visibly infected stems [48] and by reversion, in which low viral titres mean healthy cuttings can be taken even from infected

plants [49], a process that has been acknowledged for a range of virus diseases [50]. It is difficult to quantify these effects into a single parameter value; we therefore take $p = 0.8$ as a pragmatic but arbitrary default in which vertical transmission at the field scale often, but not always, occurs.

Growers' responses to differences in profit are notoriously difficult to quantify [15], and here we were forced to assume a value for our parameter describing the responsiveness of growers ($\eta$). An intuition for our default selection of $\eta = 10$ per unit of profit follows by noting that, using the default costs and losses in the "strategy vs. population" model, a controller will have an approximately 80% chance of switching to become a non-controller if there is sufficient infection in the rest of the system such that it is almost certain they will become infected over the next season (i.e. if the probability of infection of a controller, $q_C \approx 1$; Eq 17).

## 2.3 Strategies, outcomes and profits

For simplicity, we assume only a single control strategy is available, the CSS. Thus, there are only two pure strategies available to growers at the beginning of each season: to control for disease, or to not control (Fig 1B). Within each strategy, there are two outcomes realised at the time of harvesting: the grower's field may remain uninfected or have become infected.

Each strategy × outcome combination results in a different payoff (Fig 1B). An uninfected field generates yield $Y$, with loss of income, $L$, in infected fields. Control incurs a cost $\phi$. The profit for each outcome is:

$$P_{SN} = \text{Profit for non} - \text{controller who remains susceptible} = Y, \tag{11}$$

$$P_{IN} = \text{Profit for non} - \text{controller who becomes infected} = Y - L, \tag{12}$$

$$P_{SC} = \text{Profit for controller who remains susceptible} = Y - \phi, \tag{13}$$

$$P_{IC} = \text{Profit for controller who becomes infected} = Y - \phi - L, \tag{14}$$

Rational growers will never control if $L < \phi$, since then the cost of clean seed exceeds the loss due to infection. We therefore only consider $L > \phi$, from which it follows that:

$$P_{SN} > P_{SC} > P_{IN} > P_{IC}. \tag{15}$$

## 2.4 Risks of infection

We initially focus on the case in which growers can estimate instantaneous risks of infection precisely. In particular, we assume that growers use their current instantaneous probability of infection as an estimate of their instantaneous probability of horizontal infection ($p(\text{Horiz})$) over the next season:

$$p(\text{Horiz}) = \frac{\text{Instantaneous infection rate}}{\text{Instantaneous infection rate} + \text{Harvesting rate}},$$
$$= \frac{\beta(I_C + I_N)}{\beta(I_C + I_N) + \gamma}. \tag{16}$$

For controllers, clean seed rules out vertical transmission. Eq 16 therefore sets a grower's

estimate of the probability of infection next season if they were to choose to control ($q_C$):

$$
\begin{aligned}
q_C &= \text{Growers estimate of the probability of infection next season if control is adopted,} \\
&= \frac{\beta(I_C + I_N)}{\beta(I_C + I_N) + \gamma}.
\end{aligned}
\tag{17}
$$

For non-controllers, vertical infection must also be considered. We again assume growers can estimate the relevant instantaneous probability of vertical transmission ($p_N(\text{Vert})$) with perfect accuracy:

$$
\begin{aligned}
p_N(\text{Vert}) &= \frac{\text{Probability of vertical transmission}}{\text{Total number of fields}}, \\
&= \frac{p(I_C + I_N)}{N}.
\end{aligned}
\tag{18}
$$

A non-controller's instantaneous probability of horizontal infection ($p(\text{Horiz})$) must account for the fact that they may first be infected via vertical transmission, leading to:

$$
\begin{aligned}
p_N(\text{Horiz}) &= \frac{\text{Instantaneous probability}}{\text{of horizontal transmission}} \times \left(1 - \frac{\text{Instantaneous probability}}{\text{of vertical transmission}}\right), \\
&= \left(\frac{\beta(I_C + I_N)}{\beta(I_C + I_N) + \gamma}\right)\left(1 - \frac{p(I_C + I_N)}{N}\right).
\end{aligned}
\tag{19}
$$

Combining these gives us the instantaneous probability of infection for a non-controller ($q_N$):

$$
\begin{aligned}
q_N &= \text{Growers estimate of the probability of infection next season if control is not adopted,} \\
&= p_N(\text{Vert}) + p_N(\text{Horiz}), \\
&= \frac{p(I_C + I_N)}{N} + \left(\frac{\beta(I_C + I_N)}{\beta(I_C + I_N) + \gamma}\right)\left(1 - \frac{p(I_C + I_N)}{N}\right).
\end{aligned}
\tag{20}
$$

How these probabilities of infection change with increasing disease pressure ($I_N + I_C$) is shown in Fig AA in S1 Text.

## 2.5 Expected profits

Our behavioural models assume that—at the time of planting—individual growers estimate the expected profit for the next season of adopting each strategy. These depend on the payoffs for each strategy × outcome combination (Eqs 11–14) and the estimated probabilities of infection under each strategy (Eqs 17 and 20). In particular, the estimated expected profit for a controller, $P_C$, would be

$$
\begin{aligned}
P_C &= \text{Growers estimate of the expected profit next season if control is adopted,} \\
&= q_C P_{IC} + (1 - q_C)P_{SC}, \\
&= Y - \phi - L\left(\frac{\beta(I_C + I_N)}{\beta(I_C + I_N) + \gamma}\right).
\end{aligned}
\tag{21}
$$

The corresponding estimate without control, $P_N$, is

$$P_N = \text{Growers estimate of the expected profit next season if control is not adopted,}$$
$$= q_N P_{IN} + (1 - q_N) P_{SN},$$
$$= Y - L\left(\frac{p(I_C + I_N)}{N} + \left(\frac{\beta(I_C + I_N)}{\beta(I_C + I_N) + \gamma}\right)\left(1 - \frac{p(I_C + I_N)}{N}\right)\right). \tag{22}$$

The expected profit averaged over all growers, $P$, is

$$P = \text{Growers estimate of the expected profit next season averaged over the population,}$$
$$= P_C\left(\frac{S_C + I_C}{N}\right) + P_N\left(\frac{S_N + I_N}{N}\right). \tag{23}$$

Calculating $P$ therefore requires growers to know the instantaneous proportion of the population controlling. This calculation of $P$ is instantaneous, i.e. based on the state of the system at the time of planting, and provides an estimate for the next season.

## 2.6 Behavioural models

We systematically examine different ways to construct the switching terms in Eqs 5–8. Differences between behavioural models turn on how $\Delta$, an expected difference in profit (and thus profitability), is estimated. We consider four possible comparisons:

- *Strategy vs. population.* Growers compare the expected profit for their current strategy with the expected profit across the entire population of growers ($P - P_j$, with $j \in \{C, N\}$).

- *Strategy vs. alternative strategy.* Growers compare the expected profit for their current strategy with the expected profit of the strategy which they did not use ("alternative strategy"); $P_i - P_j$, with $i, j \in \{C, N\}$ and $i \neq j$.

- *Grower vs. population.* Growers compare their own outcome during the previous season with the expected profit across the entire population of growers ($P - P_G$, where $P_G$ is the grower's profit from the previous season).

- *Grower vs. alternative strategy.* Growers compare their own outcome during the previous season with the expected profit of the alternative strategy ($P_i - P_G$, where $i, j \in \{C, N\}$).

All set $z$, the probability of an individual grower switching strategy, as a function of $\Delta$, via

$$z = \max(0, 1 - e^{-\eta\Delta}) = \begin{cases} 1 - e^{-\eta\Delta} & \text{if} \Delta \geq 0 \\ 0 & \text{otherwise.} \end{cases} \tag{24}$$

Since growers aim to maximise their profit, they only consider switching strategy when the comparison appears to offer a higher payoff (i.e. when $\Delta > 0$; Fig 1C). The parameter $\eta$ sets the responsiveness of growers to a unit difference in profit.

A summary of the information required for each means of comparison is provided in Table 2.

The "strategy vs." models are inspired by evolutionary game theory [10, 25], with $\Delta$ depending on the expected profit for the strategy currently adopted. The "grower vs." models are closer to previous usage in plant disease epidemiology (e.g. [12, 13] and [17]), with the outcome obtained by the grower for their last crop used as the point of comparison. Table 3 provides a summary of the symbols used in the behavioural model.

**Table 2. Summary of the information required by growers to carry out each means of comparison.** "Controller prevalence" refers to the proportion of growers that adopted control the last time their field was planted (i.e. $\frac{S_C + I_C}{N}$) and "disease prevalence" is the proportion of growers that are currently infected (i.e. $\frac{I_N + I_C}{N}$). The "✓" refers to when information is required, and "X" when it is not necessary for decision-making.

|  | Disease prevalence | Controller prevalence | Own outcome |
|---|---|---|---|
| Strategy vs population | ✓ | ✓ | X |
| Strategy vs alternative | ✓ | X | X |
| Grower vs population | ✓ | ✓ | ✓ |
| Grower vs alternative | ✓ | X | ✓ |

**2.6.1 Strategy vs. population.** In the "strategy vs." models, the switching probabilities are decoupled from the outcome during the previous season, i.e. $z_{SC} = z_{IC} = z_C$ and $z_{SN} = z_{IN} = z_N$. The probability of a controller switching to no longer use clean seed is

$$z_{SC} = z_{IC} = z_C = \max(0, 1 - e^{-\eta(P - P_C)}). \tag{25}$$

This probability is only non-zero when the expected profit for a controller is assessed to be smaller than the expected profit over the entire population (i.e. when $P_C < P$). Conversely, the probability of a non-controller switching is

$$z_{SN} = z_{IN} = z_N = \max(0, 1 - e^{-\eta(P - P_N)}), \tag{26}$$

which takes a non-zero value only when $P_N < P$. Since $P$ is a convex combination of $P_C$ and $P_N$ (the two bracketed terms in Eq 23 are complementary probabilities), only one of $P_C$ or $P_N$ can be greater than $P$ at any time, meaning $z_C$ and $z_N$ cannot simultaneously be non-zero (although if $P_C = P_N$, i.e. the expected profits are equal, then $P = P_C = P_N$ and so $z_C = z_N = 0$).

How the expected profits and switching terms change with increasing disease pressure ($I_N + I_C$) is shown in Fig AB-C in S1 Text and Fig AE-F in S1 Text.

**2.6.2 Strategy vs. alternative strategy.** This model is similar to the "strategy vs. population" model, but now instead of comparing to the expected profit of the population, $P$, growers compare to the expected profit of the alternative strategy (i.e. $P_C$ or $P_N$, as appropriate). The probability of a controller switching is

$$z_{SC} = z_{IC} = z_C = \max(0, 1 - e^{-\eta(P_N - P_C)}), \tag{27}$$

which takes a non-zero value only when $P_N > P_C$, i.e. when non-controllers are expected to be

**Table 3. Summary of symbols used in defining the behavioural models.** The outcomes $i \in \{S, I\}$ represent—at the end of the season—fields that are (S)usceptible and (I)nfected, respectively. The strategies $j \in \{C, N\}$ correspond to (C) ontrollers (i.e. use clean seed) and (N)on-controllers (i.e. do not use clean seed), respectively.

| Symbol | Meaning | Defined in |
|---|---|---|
| $z_{ij}$ | Switching probability for grower in infection class $i \in \{S, I\}$ who used strategy $j \in \{C, N\}$ | Eqs 1–4 |
| $P_{ij}$ | Profit for grower with outcome $i \in \{S, I\}$ who used strategy $j \in \{C, N\}$ | Eqs 11–14 |
| $q_j$ | Estimated probability at time of planting of infection if go onto use strategy $j \in \{C, N\}$ | Eqs 17 & 20 |
| $p_N(\text{Vert})$ | Estimated probability of vertical infection for non-controllers | Eq 18 |
| $p_N(\text{Horiz})$ | Estimated probability of horizontal infection for non-controllers | Eq 19 |
| $P_j$ | Estimated profit at time of planting if go onto use strategy $j \in \{C, N\}$ | Eqs 21 and 22 |
| $P$ | Estimated profit at time of planting averaged over entire population | Eq 23 |
| $\Delta$ | Estimated difference in expected profit | Eq 24 |
| $q_\beta$ | Perceived value of the rate of horizontal transmission ($\beta$) | Eq 37 |
| $q_I$ | Perceived number of infected fields | Eq 38 |

more profitable next season. Conversely, the probability of a non-controller switching is

$$z_{SN} = z_{IN} = z_N = \max(0, 1 - e^{-\eta(P_C - P_N)}), \tag{28}$$

which is non-zero only when $P_C > P_N$. Here it is clear that only one of $z_C$ and $z_N$ can be non-zero at any time, since either $P_C > P_N$ or $P_C < P_N$ (or $P_C = P_N$, in which case no grower would switch since the expected profits of the two strategies are equal, and $z_C = z_N = 0$).

**2.6.3 Grower vs. population.** In the "grower vs." models, decisions are made on the basis of grower's own outcome for the last crop, and so all four switching probabilities can differ. The ordering of profits in Eq 15 means that an uninfected non-controller (i.e. class $S_N$) should never switch strategy, since these "successful free-riders" [51] obtained the highest possible profit for the last crop. Similarly, having obtained the "suckers' pay off", a grower who controlled for disease but was nevertheless infected (i.e. class $I_C$) will always have a non-zero probability of switching, since they must have obtained a lower than average profit. Whether or not $S_C$ and $I_N$ growers switch strategy depends on their profit relative to the expectation over the whole population, $P$. The switching terms are therefore

$$z_{SN} = 0, \tag{29}$$

$$z_{IN} = \max(0, 1 - e^{-\eta(P - P_{IN})}), \tag{30}$$

$$z_{SC} = \max(0, 1 - e^{-\eta(P - P_{SC})}), \tag{31}$$

$$z_{IC} = 1 - e^{-\eta(P - P_{IC})}. \tag{32}$$

**2.6.4 Grower vs. alternative strategy.** This model is similar to the "grower vs. population" model, but now growers compare to the expected profit of the alternative strategy (i.e. $P_C$ or $P_N$) rather than that of the population, $P$, with

$$z_{SN} = 0, \tag{33}$$

$$z_{IN} = \max(0, 1 - e^{-\eta(P_C - P_{IN})}), \tag{34}$$

$$z_{SC} = \max(0, 1 - e^{-\eta(P_N - P_{SC})}), \tag{35}$$

$$z_{IC} = 1 - e^{-\eta(P_N - P_{IC})}. \tag{36}$$

## 2.7 Systematic uncertainty

The models as presented thus far presume that all growers can perfectly perceive the risk of infection by knowing the true values of $\beta$ and the total number of infected individuals ($I$). However, these quantities must actually be estimated by growers. To account for this, we introduce two new parameters: the perceived rate of horizontal transmission ($q_\beta$) and the perceived number of infected fields ($q_I$). These are given by:

$$q_\beta = v_\beta \beta, \tag{37}$$

$$q_I = v_I(I_N + I_C). \tag{38}$$

Therefore $v_\beta$ and $v_I$ scale the values of $\beta$ and $(I_C + I_N)$ respectively, amounting to either an

over- or underestimate of their values. This modifies their contributions to the estimated profits in Eqs 21 and 22, which become:

$$\tilde{P}_C = Y - \phi - L\left(\frac{q_\beta q_I}{q_\beta q_I + \gamma}\right).\tag{39}$$

and

$$\tilde{P}_N = Y - L\left(\frac{pq_I}{N} + \left(\frac{q_\beta q_I}{q_\beta q_I + \gamma}\right)\left(1 - \frac{pq_I}{N}\right)\right).\tag{40}$$

Importantly, these changes to how growers perceive the risk of infection do not change the epidemiology of the system, only the behaviour of the growers. If the value of $v_I(I_N + I_C) > N$, we assume that growers estimate that all fields in the system are infected and thus set $v_I(I_N + I_C) = N$.

## 2.8 Spatial-stochastic model

We use an individual-based version of our model to test the robustness of our conclusions to spatially explicit and/or stochastic effects (see S2 Text for full details). Essentially, we modelled disease spread on a 1,000 Ha landscape containing $N = 750$ fields. Disease spreads both via movement of a whitefly vector (with the probability of a field becoming infected depending on its proximity to other infected fields) and via replanting of infected material. Decisions regarding control (which take the same functional form as the deterministic models, but now account for the dependence of the force of infection on each grower's spatial location) are made at the time of planting, which occurs asynchronously across fields. We also investigated the effect of growers' misperception in this model, focusing on the dispersal scale of the whitefly vector ($\alpha$) and the rate of horizontal transmission $\beta_S$ (Table A in S2 Text).

## 3 Results

### 3.1 Effect of profit comparison's formulation

**3.1.1 Characterising model equilibria.** The field-scale basic reproduction number in the absence of control (S3 Text) is:

$$R_0 = \text{Basic reproduction number} = R_0^H + R_0^V = \frac{\beta N}{\gamma} + p,\tag{41}$$

with distinct components corresponding to horizontal ($R_0^H$) and vertical ($R_0^V$) transmission [52] (S3 Text). This applies for both the "strategy vs." and "grower vs." models; below this threshold, there is a disease-free equilibrium (DFE) where there are no infected fields and consequently no controllers (as there is no need to control from disease where there is no risk of infection) (Fig 2A–2C). Other, disease-endemic equilibria are possible and depend on the type of comparison being used.

There are up to four different equilibria, defined as:

- **Disease-free equilibrium** at which the disease is not able to spread even when there is no control via clean seed. This is stable for $R_0 < 1$.

- **Control-free, disease-endemic equilibrium** at which the disease is endemic, but it is unprofitable to use the CSS and so no growers control.

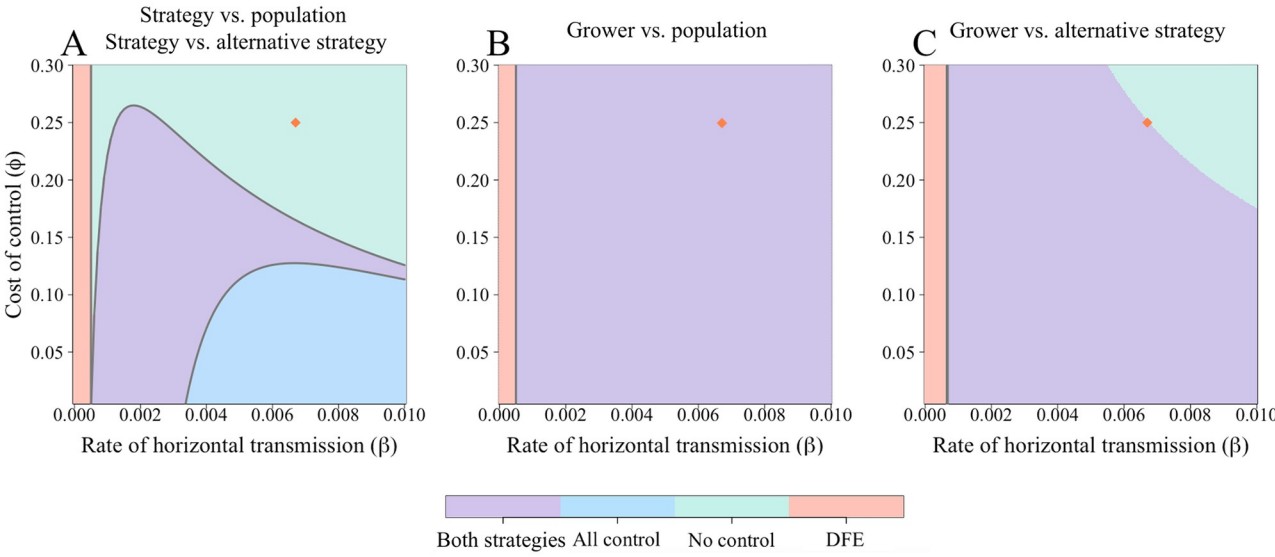

**Fig 2. Effect of the rate of horizontal transmission ($\beta$) and the cost of control ($\phi$) on final equilibrium attained.** A The "strategy vs. population" and "strategy vs. alternative" models have four possible, mutually-exclusive equilibria. For any given parameter set, both of these models will have the same equilibrium values. Which is attained depends both on $R_0$ and the balance of benefit between controlling and not controlling. Below $R_0 = 1$ (which corresponds to the vertical line at $\beta = 0.0005$ day$^{-1}$), only the disease-free equilibrium (DFE) is possible. Boundaries between equilibria are determined by the stability conditions (S3 Text). Also shown in B and C are the equilibria for the "grower vs." models, though numerical expressions determining these cannot be derived analytically. Parameters are as in Table 1; the orange diamonds mark the default values $\beta$ and $\phi$ (which, in C, is in the "no control" region). The dynamics for these default values are shown in Fig 3. B does not show all equilibria achievable for the "grower vs. population" model (see Fig A in S3 Text).

- **All-control, disease-endemic equilibrium** at which all growers control, but nevertheless disease is still endemic in the system.

- **Two-strategy, disease-endemic equilibrium** at which both disease and control equilibrate at some intermediate level.

Which subset of these outcomes are possible will depend on which model formulation is used.

Parameterisations leading to each of the four equilibria are possible for the "strategy vs." models, depending on the balance between the rate of horizontal transmission ($\beta$) and the cost of control ($\phi$) (Fig 2A). In these models, coupling of the switching terms allows mathematical analysis (S3 Text), and stability conditions can be derived for the single-strategy equilibria. As noted above, although the dynamics in reaching equilibrium can differ, the final state of the system does not depend on whether the "strategy vs. population" or "strategy vs. alternative strategy" model is considered. In each case, for an equilibrium with both strategies present, the expected payoffs ($P_C$ and $P_N$) must be equal, otherwise one group of growers will continue to be incentivised to change strategy. For only controllers to be present, $P_C > P_N$ and vice versa for the non-controller equilibria. The differences in dynamics approaching equilibrium arise from the different calculations of $\Delta$ (Eqs 25–28).

Importantly, the "all control" equilibrium, can be attained in these "strategy vs." models. This equilibrium is due to the imperfect protection provided by the control scheme; if horizontal infection is sufficiently significant, even if all growers control it does not guarantee the system remains disease-free. We note here that the equilibrium attained is independent of responsiveness of growers, $\eta$, though it does affect the dynamics approaching equilibrium (Fig AD in S1 Text).

The more complex form of the "grower vs." models makes mathematical analysis difficult, as conditions for model equilibria no longer simply depend on the difference in profits between controllers and non-controllers. However, extensive numerical work (S3 Text) indicates the equilibrium that is reached again depends only on model parameters (i.e. there is no bistability between equilibria). In the "grower vs." models, only three of the four equilibria can be attained: the "all control" equilibrium is impossible because growers who controlled but nevertheless became infected always consider switching strategy, as they are earning the lowest possible payoff ($P_{IC}$, the "sucker's payoff"). As long as there is a non-zero probability of infection (which is necessary for control to be worthwhile) there will always be non-controlling growers. Additionally, for the "grower vs. population" model, the "no control" equilibrium is only possible if all non-controllers are infected at equilibrium (S3 Text). This requires a very narrow range of parameters, shown in Fig A in S3 Text). Unlike in the "strategy vs." models, where the equilibrium is not dependent on the value of responsiveness of growers, for the "growers vs." models $\eta$ does affect the final equilibrium values (Fig A in S4 Text).

**3.1.2 Default behaviour of models.** Using the default parameters, we obtain a "no control" equilibrium when growers employ the "strategy vs. population" and "strategy vs. alternative" models (Fig 3A–3B respectively). At these parameter values, the rate of horizontal transmission and probability of vertical infection are so high that growers are likely to become infected, thus paying the cost of control ($\phi$) and the loss due to disease ($L$). Therefore, control is not worthwhile.

For the "grower vs. population" model, around 38% of growers control at equilibrium. For these parameters, any non-controlling grower with an infected field ($I_N$) has a non-zero

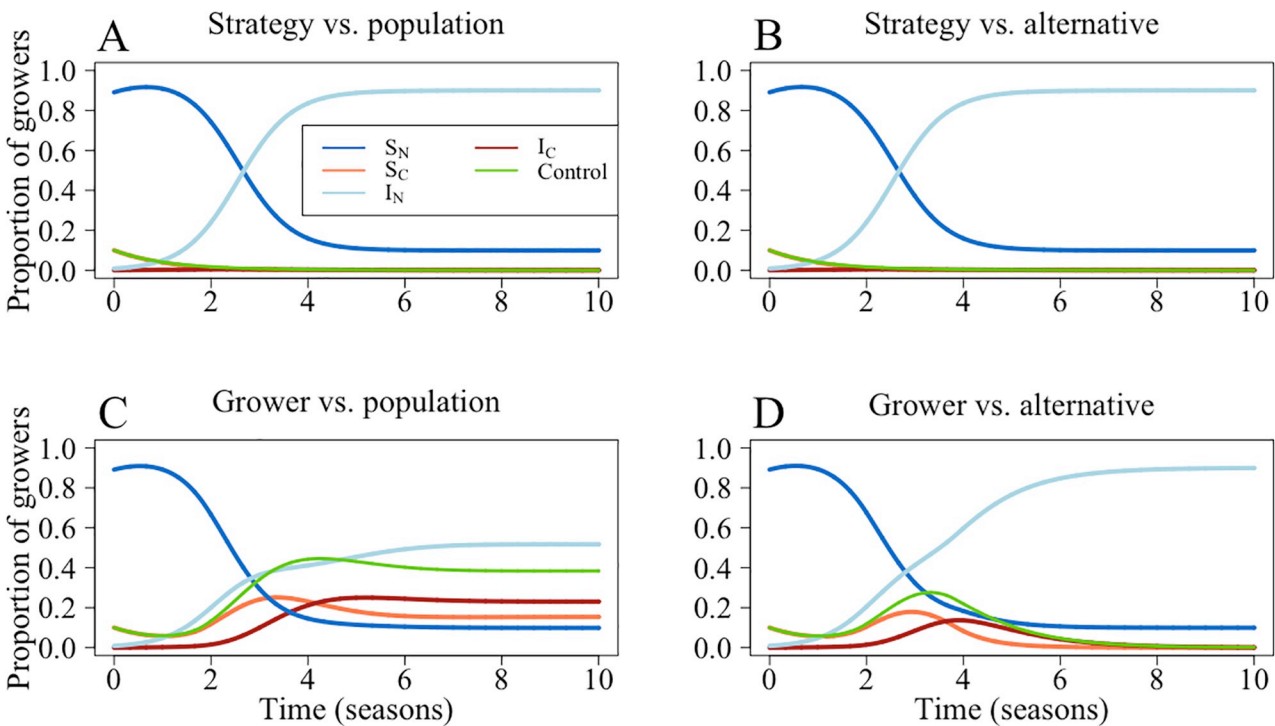

**Fig 3. Model dynamics for different means of comparison.** Here, $S_N$ are susceptible non-controllers, $I_N$ infected non-controllers, $S_C$ are susceptible controllers and $I_C$ infected controllers. $C$ represents $S_C + I_C$. In A, growers compare the expected profit of their strategy with that of the entire population; in B they compare the expected profit of their strategy with the expected profit of the alternative strategy. For C, growers compare their own outcome with expected profit of the population and in D they compare their outcome with the expected value of the alternative strategy. In A, B and D, no growers control for disease at equilibrium, though in C around 38% of growers control. Parameters are outlined in Table 1.

probability of changing strategy as the presence of uninfected non-controllers ($S_N$) means that they will earn below the expected population profit, $P$ (see Eq 30). Thus, there will be controllers at equilibrium. Only when the parameter values mean that there will be no susceptible growers will there be a "no control" equilibrium (Eq 56).

This is not observed in the "grower vs. alternative" model, where no growers control at equilibrium (Fig 3D). For this model, infected non-controlling growers ($I_N$) need only earn more than the alternative strategy, so if there is a high number of infected controllers $I_N$ should not switch strategy (Eq 55).

## 3.2 Effect and implications of epidemiological and economic parameters on uptake of control

We now continue our investigation using only the "grower vs. alternative" model set-up. The idea of the "reflexive producer" means that growers are likely to respond to previous season's outcomes [18], but still incorporate information about what the likely infection risk is for the next season [20, 53]. We therefore believe that this is a useful approximation of how growers may actually perceive profitability, though we include analysis of the other models in Fig C in S4 Text.

**3.2.1 Effect of epidemiological and economic parameters on uptake of control.** We investigate the change in uptake of control and proportion of disease fields when the cost of control ($\phi$), probability of vertical transmission ($p$) and loss due to disease ($L$) are varied (Fig 4). Responses to changes in parameter values were often intuitive. For example, at very high costs of control ($\phi$), fewer growers controlled for disease (Fig 4A). Similarly, when the risk of vertical transmission increased, so too did the proportion of growers controlling (Fig 4B). However, even within these broad trends, there were also non-monotonic responses. We found that at medium-to-low values for cost of control ($\phi$) and the rate of horizontal infection ($\beta$), high proportions of growers controlled for disease (Fig 4A). Yet as $\beta$ increased, the higher probability of infection narrows the range of costs for which a controller will consider participation in the CSS, as the grower will likely have to pay both the cost of control ($\phi$) and the loss due to disease ($L$) (the "sucker's payoff").

A similar principle underpins Fig 4B–4C. At higher probabilities of horizontal transmission ($\beta$), disease pressure increases and controllers are likely to be infected, even if non-controllers are more likely to be infected due to high probabilities of vertical transmission ($p$). In each case, when there is a higher proportion of controllers at equilibrium, there are fewer infected fields (Fig 4D–4F).

Growers receive the highest profits when $R_0 < 1$ (i.e. there is no disease, so growers cannot incur any yield loss) (Fig 5). When disease is endemic to the system (i.e. $R_0 > 1$), both profit and yield decrease with higher costs, losses and probability of infection (corresponding to the higher proportion of infected fields in Fig 4D–4F). The lowest yield and profit are obtained when all fields are infected, and at this point no-one controls for disease.

**3.2.2 Implications for encouraging adoption of control.** Growers are responsive to both economic and epidemiological factors. If the cost of the control mechanism is too high, growers will not use the scheme as they do not perceive it to be worthwhile. If the scheme provides too little a benefit over the non-control option (which in this case would mean that the probability of vertical transmission was relatively low), growers will also not use it. Though lower losses are beneficial for profit and yield, they discourage disease control. If losses are sufficiently low (such that $L < \phi$), control would not be a rational choice. It is therefore important to emphasise the importance of subsidies for control schemes (which will have the effect of reducing $\phi$) to policy-makers in order to encourage participation. The effect of such a subsidy

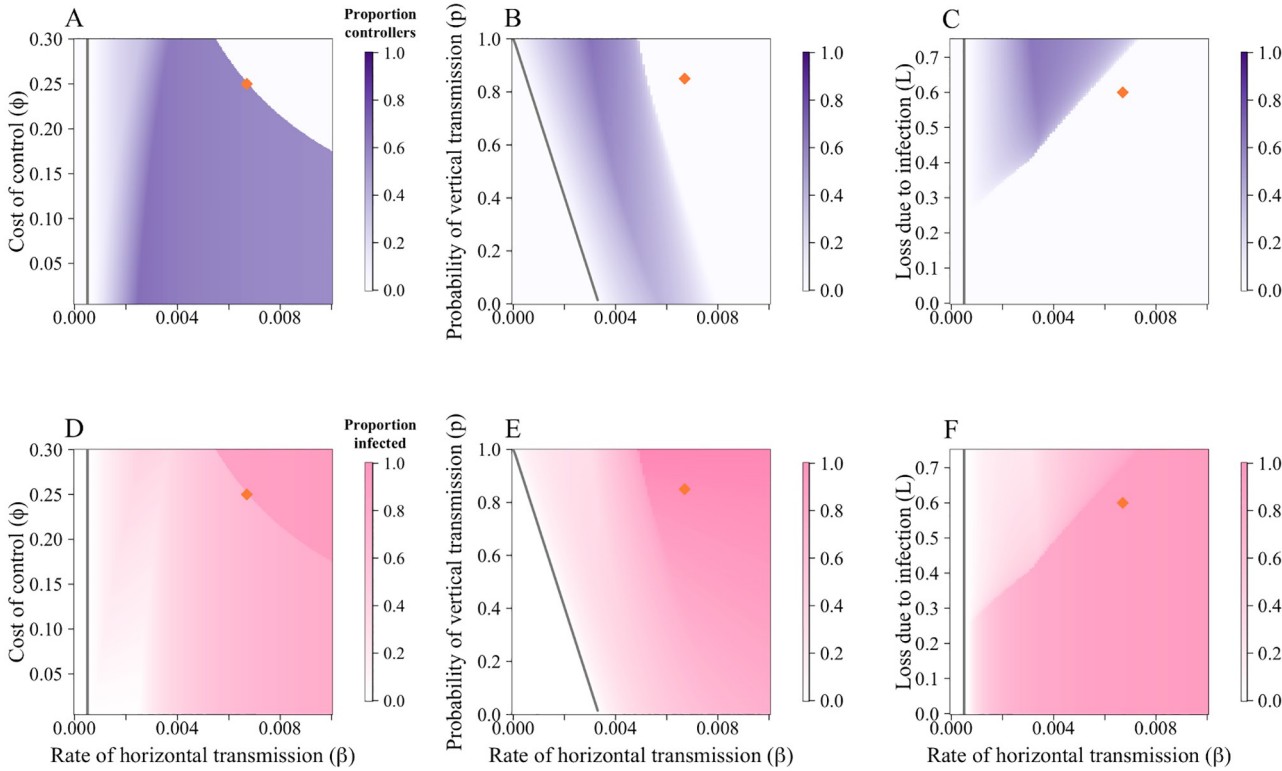

**Fig 4. CSS participation and proportion of infected fields are dependent on economic and epidemiological parameters.** Plots show the equilibrium proportion of controllers in response to variations in parameters when comparisons are made using the "grower vs. alternative" comparison. A Effect of the rate of horizontal transmission ($\beta$) and cost of control ($\phi$). B Effect of the rate of horizontal transmission ($\beta$) and probability of receiving infected planting material via trade ($p$). C Effect of the rate of horizontal transmission ($\beta$) and loss due to infection ($L$). At these default parameter values (particularly the high cost of control), both $p$ and $L$ only affects participation in the CSS for a narrow range of $\beta$. D-F show the response of the proportion of infected fields ($(I_N + I_C)/N$) to $\beta$, $\phi$, $p$ and $L$ respectively. In each case, as control increased, disease decreased. Parameters values are in Table 1; the orange diamonds mark the default values.

is shown in Fig D in S4 Text, where the lower cost of control widens the values of the rate of horizontal transmission ($\beta$) and the probability of vertical transmission ($p$) for which control is profitable.

## 3.3 Effect of systematic uncertainty

Even when growers underestimate the rate of horizontal transmission, $\beta$, to the extent that they do not believe there to be any risk of horizontal transmission (i.e the relative perception of $\beta$ is zero ($\nu_\beta = 0$, so the perceived value of $\beta$ ($q_\beta$) = 0), some still use the CSS as the probability of vertical transmission is non-zero (Fig 6A). As $\nu_\beta$ increases, fewer growers use the control scheme as they perceive they will achieve the "sucker's payoff". The default parameterisation (when $q_\beta = \beta$, i.e. $\nu_\beta = 1$) means that growers should never control. The higher cost of control ($\phi$) caused the system to reach a "no control" equilibrium Fig 6A).

Unlike when misestimating $\beta$, misestimations of $I$ affect perception of both horizontal and vertical transmission. This means there is a less straightforward response to misestimations of $I$. The kinks in (Fig 6B) are caused by changes in the values of the switching terms (Eqs 33–36), which set the probability that a grower managing a field of a particular type will switch strategy.

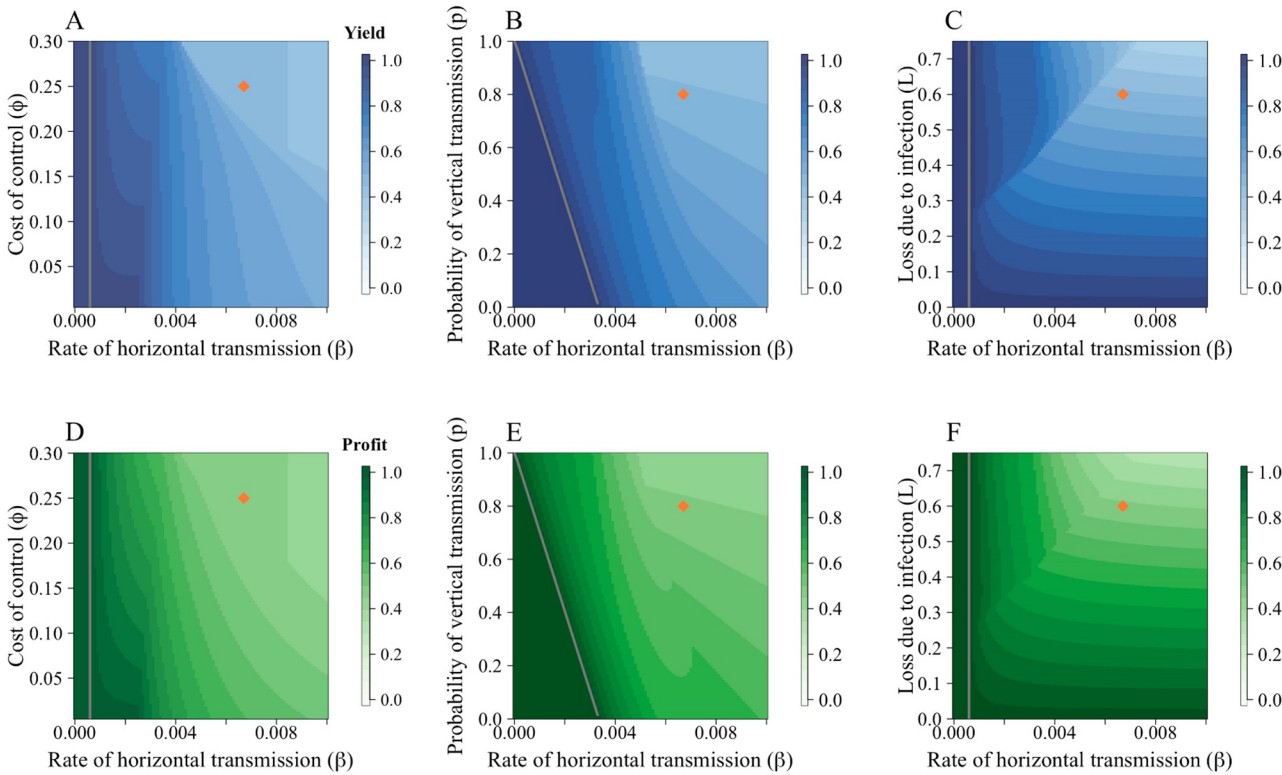

**Fig 5. Effect of parameters on yield and profit for the "grower vs. alternative" models.** A—C show the average yield, whilst D—F show the average profit. For each parameter scan, at low rates of horizontal transmission ($\beta$), average yield and profit was higher as very few fields were infected (and thus did not incur the loss due to disease, $L$). A The effect of the cost of control, $\phi$, on the yield. As the cost of control increases, fewer growers participate in the CSS and disease incidence increases, leading to lower yields. B The effect of the probability of vertical transmission, $p$. At higher values of $p$, yield falls as disease prevalence increases. C Effect of loss due to disease, $L$. Intuitively, at higher $L$, yield falls. Parameters are as in Table 1; the orange diamonds mark the default values. Patterns were similar across D—F, with lower profits observed as $\phi$, $p$ and $L$ increased.

When the relative perception of $I$, $v_I$, is small and increasing, there are more controllers at equilibrium (Fig 6B). At low values of $v_I$, the non-infected controllers ($S_C$) growers perceive that the profit for non-controllers is greater than what they have earned ($\tilde{P}_N > P_{S_C}$), so they should still consider switching strategy (i.e. $z_{S_C} > 0$) (Fig 6C). As $v_I$ increases, their perception of $\tilde{P}_N$ falls and fewer $S_C$ growers switch. This causes an increase in infected controllers ($I_C$), fewer $S_C$ switch strategy before they become infected. Eventually, $\tilde{P}_N < P_{S_C}$ and $S_C$ growers stop switching strategy entirely (around season 4 in Fig 6D). Many of these controllers will then become infected and switch strategy, leading to lower participation in control with higher $v_I$. As $v_I$ increases, $I_N$ growers are less likely to switch strategy as $\tilde{P}_C$ falls. Eventually, $z_{IN} = 0$ and no $I_N$ growers should switch strategy (Fig 6F). As all $I_C$ growers should have a non-zero probability of switching strategy, but no non-controllers should switch, this leads to a "no control" equilibrium.

## 4 Discussion

Surprisingly, few plant disease studies examine how human behaviour alters the dynamics of epidemic models. Here we presented a model that uses game theory to incorporate grower decision-making into a model of CBSD and investigated the effect of economically- and epidemiologically- important parameters on disease spread and the uptake of control by way of a

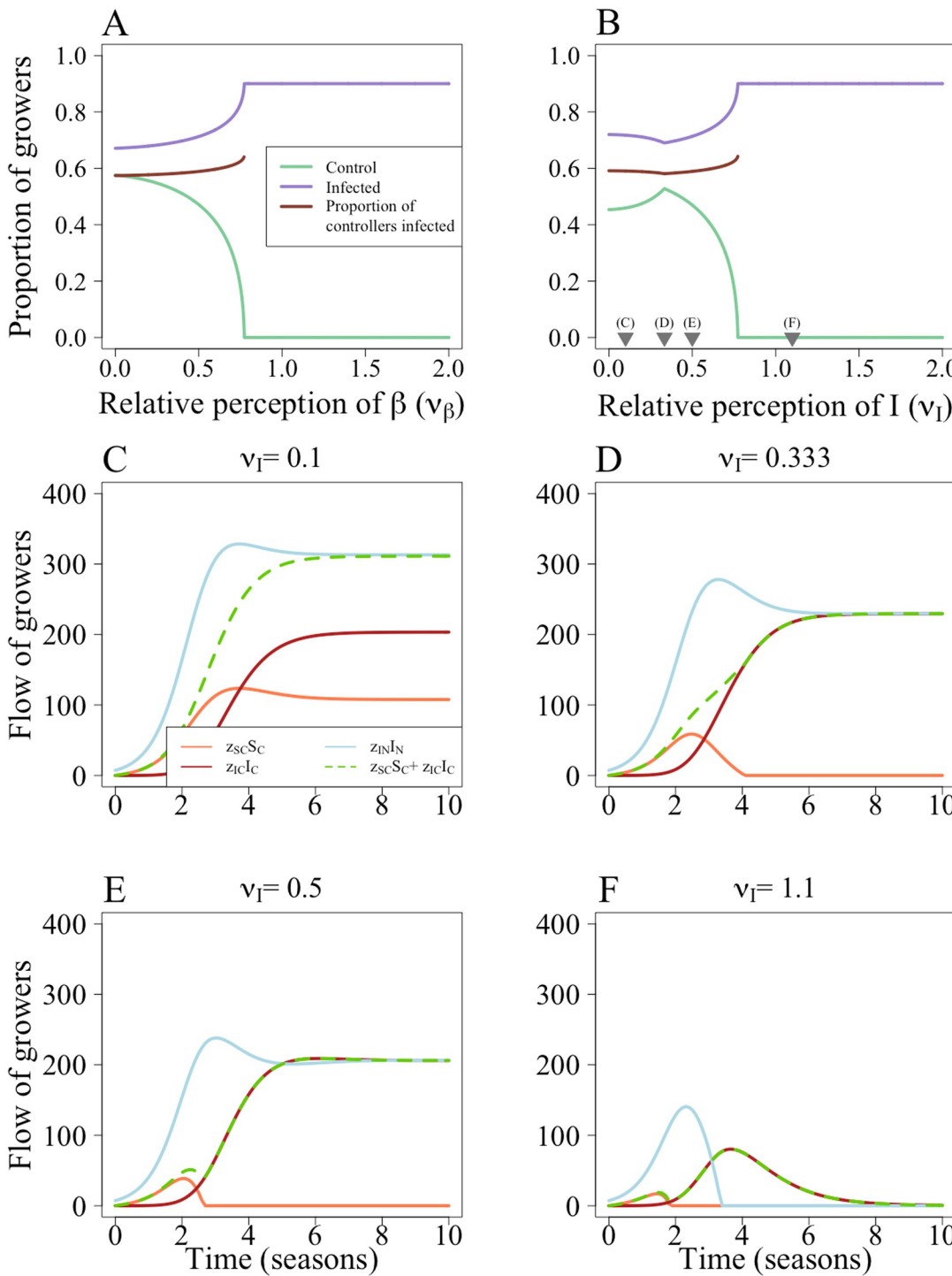

**Fig 6. Effect of systematic uncertainty of epidemiological parameters and quantities.** When the perception of $\beta$ ($v_\beta$) is small there is a higher proportion of controllers A, which decreases as $v_\beta$ increases. In B, there is initially an increase in the proportion controlling with the perception of $I$ ($v_I$). In B the number of infected fields levels out once the conditions for a "no control" equilibrium are met. The grey triangles in B show the values of $v_I$ used in C-F. C When $v_I = 0.1$, $S_C$ growers still perceive that it is profitable to change strategy ($z_{SC} > 0$). D At $v_I = 0.33$, as the number of infected fields increases, the perceived value of $P_N$ ($\tilde{P}_N$) decreases and $z_{SC} = 0$. However, they continue to get infected and there is an increase in $I_C$ fields. E At $v_I = 0.5$, $S_C$ fields stop switching strategy earlier in the epidemic. As $\tilde{P}_C$ falls, so does $z_{IN}$. F At $v_I = 1.1$, $\tilde{P}_C < \tilde{P}_{IN}$, so $z_{I_N} = 0$. This leads to a "no control" equilibrium. Parameters are as in Table 1.

clean seed system. We found that the formulation of the model (whether based on rational or strategic-adaptive expectations) had a significant consequences on the possible equilibria that could be attained. The model parameters also influenced outcomes, and interactions between them (particularly the cost of control and the rate of horizontal transmission) had large impacts on growers' participation in the CSS. Finally, the response to misperception of epidemiological quantities was often not intuitive, and could have implications for the communication of risk.

We found that basing decisions on rational (the "strategy vs." models) or strategic-adaptive ("grower vs." models) changed the potential outcomes for the model. Only the "strategy vs." comparisons permitted an "all control" equilibrium, whereas the low payoff for controllers with infected fields (the "sucker's payoff") meant that they will always have a non-zero probability of switching strategy in the "grower vs." comparisons. This difference in possible equilibria from the models shows that it is important not only to include behaviour in epidemic models, but to consider how it is included based on what quantities are being compared and what information will be available to individual growers, such as the disease pressure, profit of other growers and breakdown of the expected profit for each strategy.

Game theory predicts that high participation in voluntary disease control schemes is difficult to attain, as the success of disease control policies in reducing the probability of infection disincentivises continued use of the scheme [22, 25, 54]; reviewed in [7]. Indeed, in the case of voluntary vaccinations, at least with perfect immunity, models predict that actual level of vaccination will be less than that needed for herd immunity [2]. This is because of the *positive externalities* generated by having high levels of vaccination; if the majority of individuals are perfectly protected from infection, then there will be a lower probability of infection for others who are not vaccinated. They can then get much of the benefit of vaccination without themselves incurring any risk or cost. Indeed, the risk of other individuals "free-riding" off an individual's control mechanism may also be a disincentive to control [53].

Yet if vaccines are not completely effective, the reduction in magnitude of positive externalities can lead to increased vaccine uptake until a stable "all vaccinated" equilibrium is attained [55]. That is, as the vaccine no longer confers perfect immunity to the vaccinated, they may then transmit the disease to the unvaccinated. This decreases the benefit of high levels of vaccination in the population to the unvaccinated, and thus encourages them to be engaged in disease prevention themselves. This is more akin to the control system we have considered here, where the CSS only reduces the initial probability of infection and does not provide continued control during the season.

In our model, it was the imperfect protection offered by the CSS that allowed the "all control" equilibrium to be attained in the "strategy vs." models (Fig 2A). Although the CSS eliminates a grower's chance of receiving infected material via vertical transmission, it offers no protection from infection via the whitefly vector. By comparing the expected payoff of everyone using their strategy, rather than basing decisions on their own outcomes, even growers who themselves received the lowest possible payoff do not necessarily have a positive probability of switching strategy.

In the "strategy vs." models, we found that use of CSS could reduce CBSD spread, but never lead to disease elimination. By considering the stability of the equilibria, we can see that for an "all control" equilibrium, disease must be capable of moving via the whitefly vector alone. Fields that were initially planted with clean seed can therefore become infected, allowing disease to remain endemic in the system. Thus, though increased use of the CSS leads to lower disease prevalence, some degree of disease spread is required for growers to consider the CSS worthwhile. The threshold that determines the benefit of the CSS is determined by the stability conditions outlined in S3, which are broadly determined by the probability of incurring the

loss due to disease for both controllers and non-controllers. This result, and associated requirement for horizontal transmission, holds true for the "grower vs." models, too, though analytical expressions cannot be derived to describe the stability of the equilibria.

We assumed that all growers had the same responsiveness (our parameter $\eta$). This impacted the final equilibrium value attained in our "grower vs." models (Fig A in S4 Text), but not our "strategy vs." models (Fig A in S1 Text). We have also assumed that growers would always choose the most profitable strategy; if the alternative strategy has a higher expected profit, growers have a non-zero probability of switching (Fig 1C). As $\eta$ increases, growers are almost guaranteed to switch strategy. For the "strategy vs. models", which are based on rational expectations, this increase in $\eta$ approaches the assumption of perfect rationality in game theory [24]. Both rationality and responsiveness are likely to vary from grower to grower. Their distribution can be described by a "risk attitude" framework, in which individuals fall on a spectrum between risk tolerant and risk averse [56]. Including such heterogeneity in risk perception in future studies would allow for greater understanding of behaviour in uncertain environments.

When conducting our further analysis into the effects of parameters and misperception, we focused on the "grower vs. alternative" evaluation, where individual growers compare their own profits with the expected profit of the alternative strategy. The reasoning behind this was twofold: first, this has been used previously in plant epidemiological literature [12, 13, 17]. Secondly, and most importantly, we believe this comparison to be most similar to what real growers might be expected to do [18, 20, 53]. It requires a moderate amount of information, as it only relies on the individual knowing their own profit from the previous season and the expected profit of the alternative strategy (Table 2).

Epidemiological and economic parameters had a substantial impact on the uptake of control (Fig 4). When the cost of control ($\phi$) was low, more growers used the CSS (Fig 4A and Fig D in S4 Text). However, even at these low costs of control, if the rate of horizontal transmission was too high, growers would still not use the control scheme as the high probability of infection meant they were likely to incur the dual penalty of yield loss and cost of control. Subsidies, then are likely to be an effective way of encouraging control, particularly as both profits and yields are highest when the cost of control is low (Fig 5A and 5C) In practice, the success of subsidies may be limited by a lack of information amongst growers, unwillingness of growers to vary from current practices, or perceived ineffectiveness of control schemes. It is important that the introduction of subsidies control is accompanied by outreach and educational campaigns for growers [57].

For control to be worthwhile, it must provide sufficient benefit compared to the alternative non-control strategy. In our case study, the benefit of control is the elimination of vertical transmission for those that use the CSS. Thus, if the probability of vertical transmission is sufficiently low for non-controllers, there may not be enough incentive to control (Fig 4B). Similarly, if growers do not stand to lose much yield when infected, the cost of control may be a disincentive, particularly if the probability of infection is low (Fig 4C). Indeed, if the expected yield loss over a season is less than the cost of control, it would never be worthwhile to control. It is therefore important to emphasise the benefits of control to growers, particularly if there is likely to be a significant future increase in disease prevalence that they are not accounting for when they are making their decisions.

Growers' inability to properly perceive epidemiological parameters had varying effects on CSS participation. Underestimating the rate of horizontal transmission, $\beta$, increased participation as growers believed that infection via this route was unlikely (Fig 6A). However, there was still a high probability of vertical transmission so control was worthwhile. As estimates of $\beta$ increased, control was less favourable and participation decreased. A similar trend was observed in the when estimating $\alpha$, the whitefly's dispersal scale parameter in a more complex

spatial-stochastic formulation of our model (Fig C in S2 Text). Misestimations of the proportions of infected individuals had a non-monotonic pattern, with the proportion of controllers initially increasing with the perceived number of infected fields ($q_I I$) before falling as more controllers become infected and thus receive the lowest payoff (Fig 6B).

It is therefore important to understand how growers perceive their probability of infection, as this will influence their participation in control schemes and subsequent spread of disease. Overestimating (or exaggerating when communicating with growers) the threat can potentially discourage participation due to the perceived inevitability of infection regardless of any actions taken to prevent it. The effect of underestimation depends on the route of disease transmission in question, but control only remains attractive if it is seen as providing sufficient benefits. Again, here every grower has the same level of misperception, but in reality this will be closely linked to an individual grower's attitude towards risk, access to information and previous history of infection, which all will be highly variable between growers.

We have only considered economic incentives for grower participation in the CSS. Realistically, other considerations—such as preference for local varieties, flavour, *etc.*—will play an important role in determining participation. Indeed, surveys of cassava growers in Sub-Saharan Africa have found that such qualitative traits have a greater role in determining a grower's preference than economic traits such as yield [14, 58]. These private preferences or perceptions of risk are difficult to assess and could be encapsulated in a "cost of control" parameter that will vary from grower to grower.

As well as economic considerations, there are many social factors underpinning the adoption of a new control mechanism. Descriptive norms (what the majority of people are doing, and thus what is deemed socially acceptable; [59]) have been found to influence growers' decision to partake in control [60]. Relatedly, a grower's social network and ability to access trusted information has a strong impact on management practices. Some growers will treat the advice of "experts" with a degree of scepticism (often due to a perceived conflict of interest or disconnectedness from the "true" needs of growers) [19]. Other grower have developed a strong dependency on the advice of these experts [19]. Such contrasting beliefs and degree of trust in knowledge sources, even in adjacent landscapes, highlights the heterogeneous information network used by growers. In all cases, growers placed high value on personal and trans-generational knowledge [19]. Growers may also be reluctant to adopt a new technology for fear of failure, or as they contradict a pre-existing ideology [19].

Kaup introduces the idea of a "reflexive producer" as a grower that must make decisions by balancing knowledge generated by local growers and knowledge provided by external experts [18]. This is similar to the hypothetical scenarios we previously outlined, particularly that of the "grower vs. alternative" scenario where a grower is using a particular control strategy and is then approached by an extension worker that proposes an different strategy. Thus, motivations and considerations of a grower to partake in specific control schemes are highly varied and depend on the market demands, information network available to growers, disease pressure and proximity to previous outbreaks [20].

For simplicity, in both our stochastic and deterministic models we included an average loss of yield. In reality, yield will depend on how far into the growing season the field was infected and the rate of within-field disease spread (as well as many other factors, such as environmental stochasticity, the variety of cassava in question, availability of technology, planting density *etc.* [61–63]). The inclusion of time-dependent yield loss will favour clean seed use, as these fields will on average be infected later in the growing season than conventional fields. Indeed, this could shift the balance of payoffs such that the payoff for an infected field planted with clean seed is no longer the lowest possible payoff. If this shift were observed, an "all control" equilibrium may now be stable in the "grower vs." models that we have described.

Though we have not investigated it here, CSS can "bundle" desirable traits by providing CBSD resistant or tolerant planting material, changing the nature of the externalities generated by clean seed use. There are few such varieties available for cassava. Provision of resistant material, though of obvious benefit to the recipient, will also increase the benefit experienced by free riders by reducing their probability of infection. Cassava varieties that are tolerant to CBSD infection will benefit the focal grower, but may increase the disease pressure experienced by other growers and thus incentivise control. These are potentially important distinctions, which we intend to return to in future work.

The inclusion of grower behaviour alters disease dynamics, as the option to participate in a control scheme lowered the probability of infection and has the potential to increase the yield achieved by the growers. We have found that the means of inclusion of behaviour is also important; the quantities of comparison impact dynamics approaching equilibrium, but also determine the nature of the equilibria that can be achieved. Our investigation has been limited to one case study, but the decision modelling framework laid out in this paper is sufficiently flexible to be incorporated into a range of other disease systems. It would also allow grower behaviour to feed into other types of study, for example those in which control has a spatial element and/or is done reactively in response to detected infection [64–67], or where estimates of parameters become more accurate over time and as disease spreads [68].

## Supporting information

**S1 Text. Probability of infection, responsiveness, profits, and switching terms in the "strategy vs." models.** We dissect the behaviour that drives the dynamics of the "strategy vs." models, specifically the response to the change in the number of infected fields. **Fig A: Probability of infection, responsiveness of growers, expected profits, and switching terms for the "strategy vs. population" model.** A For the default parameterisation, the probability of infection increases as the number of infected growers increase ($q_N$ and $q_C$), though the probability non-controllers will be infected via horizontal transmission ($p_{N(\text{Horiz})}$) falls as $I$ increases. B The expected profits for controllers, non-controllers and the population for different proportions of controllers in the population ($c$). For the default parameters, $P_C < P$ for all values of $I$. C Switching probabilities for controllers and non-controllers for different $c$. As $P_C < P$, controllers should always have a non-zero probability of switching strategy, whereas non-controllers should never start to control. D The responsiveness of growers ($\eta$) does not affect the final equilibrium values, though it does impact the time it takes to reach equilibrium. E shows the expected profits for controllers, non-controllers and the population when the cost of control $\phi$, = 0.125. Now, at high levels of $I$, it is profitable to control. Where $P_C = P_N$, both strategies will be present at equilibrium. F For $\phi = 0.125$, for low values of $I$ controllers should switch strategy, but as $I$ increases non-controllers should have a non-zero probability of switching into the clean seed system. For this parameter set, the equilibrium value of $I$ is therefore 411, with 349 infected controllers and 63 infected non-controllers.
(PDF)

**S2 Text. Details of the spatial-stochastic analogue of the "grower vs. alternative" model.** A comparison of the spatial-stochastic and deterministic models show good agreement between the predictions made by each model. **Fig A: Comparison of default behaviour of the "grower vs. alternative" spatial-stochastic and deterministic models.** A Dynamics for spatial model. As with the deterministic model C, under the default parameterisation no growers use the CSS after 10 seasons. Adding a subsidy in B and D allows for the two-strategy equilibrium. The figures show the mean for 100 runs of each model, and the error bars show one standard deviation. The equilibrium values and dynamics for spatial (A and C) and non-spatial models (B

and D) are very similar in both cases, emphasised in E and F, which show the proportions controlling and infected for the spatial-stochastic ("stoch.") and deterministic ("det.") models. **Fig B: Response to changes in the rate of horizontal transmission and cost of control for the "grower vs. alternative" models.** A and C The proportion of controllers and infected fields after 50 seasons for the spatial-stochastic model and B and D the equilibrium values of control ($S_C + I_C$) and infection ($I_N + I_C$) for the deterministic model. Aside from the parameters being scanned over, the default parameters are used (Table 1 and Table A in S2 Text). The results in A and C closely align with the equilibrium values in the non-spatial model (B and D), indicating that our results are robust to spatial and stochastic effects. In A and C the means over 100 runs and the error bars show one standard deviation around the mean. **Fig C: Effect of systematic misestimation in the spatial-stochastic model.** A For the default parameters, as the perceptions of the dispersal scale for the whitefly vector ($v_\alpha$) increase, fewer growers use the control scheme as they estimate that they would likely end up infected. B As perceptions of the rate of horizontal transmission increase ($v_\beta$), fewer growers use the CSS in a pattern that matches that seen in Fig 6A in the main text. The mean values were calculated over 100 runs are the error bars show one standard deviation around the mean. **Fig D: Spatial spread of infection in spatial-stochastic model.** To emphasise the spatial component of infection, vertical transmission has been removed from the model (i.e. $p = 0$). Additionally, we have removed the "control" strategy from the growers as, without vertical transmission, this would have disappeared from the population within 5 seasons. The light blue dots show infected fields, whilst the grey dots are uninfected fields. Table A: Summary of parameter values and initial conditions required for the spatial-stochastic model.
(PDF)

**S3 Text. Mathematical analysis of the "strategy vs." and "grower vs." models.** The "strategy vs." models allowed more detailed mathematical analysis, whilst the stability of equilibria in the "grower vs." models had to be determined numerically. **Fig A: Possible equilibria for the "grower vs. population" model when $p = 1$.** With the higher probability of vertical transmission, the "no control" equilibrium is possible for the "grower vs. population" model as there will be no non-infected, non-controlling ($S_N$) growers at equilibrium (Eq 56). However, now that $p = 1$, there can never be a disease-free equilibrium for this parameter set (as $R_0 > 1$). **Table A: Range of values used for parameters when evaluating the stability of the "grower vs." models.**
(PDF)

**S4 Text. Supplementary results across all deterministic models. Fig A: Effect of changes in responsiveness and horizontal transmission on the proportion of controllers at equilibrium.** Below a certain threshold, there is a decrease in the proportion of controllers with an increase in $\eta$, as the higher responsiveness causes more $S_C$ growers to switch strategy. Above this threshold, an increase in $\eta$ causes an increase in controllers. The parameter values mean that $S_C$ growers no longer change strategy, so they cannot leave the CSS. However, an increase in $\eta$ means that $I_N$ growers have a higher probability of switching into the CSS. The solid vertical line denotes where this threshold is crossed ($\beta = 0.03301$ day $-1$). **Fig B: Effect of responsiveness ($\eta$) on the "grower vs. alternative" model.** A The flow of growers between the non-infected controllers ($S_C$), infected controllers ($I_C$) and infected non-controllers ($I_N$) based on their probability of switching strategy ($z_{SC}$, $z_{IC}$ and $z_{IN}$ respectively) with $\eta = 1$. B Full model dynamics for $\eta = 1$. C The flow of growers between $S_C$, $I_C$ and $I_N$ based on their probability of switching with $\eta = 10$. Note that will lower values of $\eta$, there are fewer growers moving between strategies. D Full model dynamics for $\eta = 10$. Parameters are as in Table 1, except for $\beta = 0.004$ day $-1$, which was used to allow for an disease- and control-endemic equilibrium

(Fig 2). **Fig C: Effect of parameters on participation in the CSS.** A—C shows results for the "strategy vs. population" and "strategy vs. alternative" model formulations; D—F are for the "grower vs. population" model. Parameters are as in Table 1 in the main text; the orange diamonds mark the default values. A and D examine the impact of changing the cost of control ($\phi$) and the rate of horizontal transmission ($\beta$). At low values of $\beta$, no-one should control as the probability of infection is sufficiently low that it is not necessary. As $\beta$ increases, controlling is more beneficial and the cost of control is perceived to be worthwhile. However, as infection becomes more likely, the value of control diminishes so it is not worthwhile to invest in control. In the "strategy vs." comparisons, it is possible to reach an "all control" equilibrium, though this cannot happen for the "grower vs. population" model. B and C At very high probabilities of vertical transmission ($p$) and loss due to disease ($L$), growers will participate in the CSS, though only for a narrow range of $\beta$. However, for the "grower vs. population" models, a much wider range of all parameters allowed for participation. The irregular contours in E and F are due to oscillations around the equilibrium. **Fig D: Effect of changes in the rate of secondary transmission ($\beta$) and probability of vertical transmission ($p$) on the proportion of controllers when $\phi$ = 0.125.** Compared to the default value of $\phi$ = 0.25, a higher proportion of growers control for a broader range of parameter values.
(PDF)

## Author Contributions

**Conceptualization:** Rachel E. Murray-Watson, Nik J. Cunniffe.

**Formal analysis:** Rachel E. Murray-Watson.

**Investigation:** Rachel E. Murray-Watson, Nik J. Cunniffe.

**Methodology:** Rachel E. Murray-Watson, Frédéric M. Hamelin, Nik J. Cunniffe.

**Project administration:** Nik J. Cunniffe.

**Resources:** Nik J. Cunniffe.

**Supervision:** Nik J. Cunniffe.

**Validation:** Rachel E. Murray-Watson.

**Visualization:** Rachel E. Murray-Watson.

**Writing – original draft:** Rachel E. Murray-Watson, Nik J. Cunniffe.

**Writing – review & editing:** Rachel E. Murray-Watson, Frédéric M. Hamelin, Nik J. Cunniffe.

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
