## [Decision Letter · Decision Letter 0]

2 Mar 2022

Dear Ms Murray-Watson,

Thank you very much for submitting your manuscript "It ain't what you do, it's what drives why you do it: how the basis of growers' decision making affects the effectiveness of disease control" for consideration at PLOS Computational Biology.

As with all papers reviewed by the journal, your manuscript was reviewed by members of the editorial board and by several independent reviewers. In light of the reviews (below this email), we would like to invite the resubmission of a significantly-revised version that takes into account the reviewers' comments.

In addition to the recommendations formulated by all three reviewers regarding the reorganization of the main content of this manuscript, the title and first sentence should emphasize stronger that this work addresses only plant diseases.

We cannot make any decision about publication until we have seen the revised manuscript and your response to the reviewers' comments. Your revised manuscript is also likely to be sent to reviewers for further evaluation.

Sincerely,

Claudio José Struchiner, M.D., Sc.D.

Associate Editor

PLOS Computational Biology

Tom Britton

Deputy Editor

PLOS Computational Biology

In addition to the recommendations formulated by all three reviewers regarding the reorganization of the main content of this manuscript, the title and first sentence should emphasize stronger that this work addresses only plant diseases.

Reviewer's Responses to Questions

**Comments to the Authors:**

Reviewer #1: Congratulations to the authors on writing a coherent and interesting paper on an area that is in need of new ideas and robust methodology. I think this is an important contribution to the interdisciplinary literature. In a paper of this technical complexity it is not possible to check in detail all of the mathematics and computational results. As far as possible in the time available I tracked the argument developed through the equations provided. This somewhat superficial evaluation seemed to accord with the results offered. Work of this type tends to be fully evaluated only after publication. I can see no technical reasons to deny publication, based on my reading.

Lines 32-35. It would be good to have a citation that demonstrated that these intuitions are, in fact, correct; both those relating to disease prevalence and cost, and those relating to the actions of other growers. Recent work on HLB by Garcia Figuera et al. might be useful in this context, but since this is work from my group there is a conflict of interest in me promoting it. However there may be other works in the reference lists of these papers that would be useful. In any case, I would be inclined (if I was writing the current paper) to say that these are underlying assumptions of the model. They are not unreasonable assumptions to make of a rational agent.

https://doi.org/10.1007/s12571-020-01133

https://doi.org/10.1094/PHYTO-12-20-0544-R

Line 43-44. “Yet if too many growers engage in free-riding behaviour, there will be a resurgence of infections, making disease control more likely” Perhaps, but empirical studies (see above, and also work by Ariel Singerman in Florida) find that growers report one of the main reasons they do not participate in collective action is that they think others will free ride. Once disease starts to spread they point to the uselessness of further collaboration and the resulting lack of coordination leads to a diversity of responses; it’s not safe to assume that more disease results in more effort to reduce disease.

Lines 53-62. I’m very much in favour of pursuing a game-theoretic approach in studying this kind of problem, but this paragraph highlights one of the issues. Most growers (at least in the systems with which I’m familiar) have only a vague idea about each others’ or the industry average costs. The authors acknowledge this unrealistic assumption on line 61. Whether it’s an important issue will depend on what role games with this type of information structure play in the subsequent analysis.

Lines 63-69. This paragraph is confusing. It is not explained in advance what is meant by “the alternative strategy”. An introductory sentence before Line 53 explaining the overall approach would be useful. I would reword the last sentence (lines 68-69) to make it clearer that you mean any games of the type that compare the growers’ strategy with another strategy; perhaps just include a placeholder after the vs. inside the quotation marks “grower vs. X”?

Table 2. (Line 170) In the table heading, include a remark about whether the tick or cross mean the information is needed. It’s not difficult to work out, but give the reader a break.

Line 207. insert “s” on the end of depend to match tense.

Line 301. Italics needed for Bemesia tabasci Is this the first full mention? If not then shorten to B. tabaci

Line 337. add a reminder of what eta is to help with interpretation of the description of the dependence

Lines 347-351. I’m not completely happy with the way you word the game’s mechanism for switching here. In the model, the mathematics ensures that the equilibrium solution contains a mix of strategies, but you stray into couching things in terms of what an actual human agent might do – “considers changing strategy” – and you’re not representing a process at that level of detail. I sympathize. It’s hard to construct this kind of model without attaching human attributes to the mechanisms, but perhaps this is one step too far that direction?

Line 358. I don’t think you’re measuring the pathogen’s R in this work. Your results all relate to fields as the unit of measurement so your estimates of R_{0}R

0

relate to diseased fields, no?

Line 438-440. This is an important (if intuitively obvious) result. In driving/leading a population towards adoption of control this would be a central extension message. It would also be an important message to take to policy-makers about how funding incentives for control would be worthwhile. It’s important that the work reached this conclusion, because (earlier comments about avoiding over-interpreting the models in human terms notwithstanding) the models make relatively mild and rational assumptions about choice.

Figure 9. Vertical axis represents a number but is labeled as a proportion.

Lines 510-519. The comparison of outcomes predicted by Game Theory approaches with reality is interesting. It perhaps says as much about the limitations of the approach as it does about human nature? If the comment that the actual level of vaccination achieved will be less than that required for herd immunity is important to the rest of the paper it needs to be spelled out a bit more; it’s not intuitive, at least to me, why this should be a general result. The argument advanced in lines 515-519 is also not convincing. At least in the USA, one of the most commonly offered arguments against getting vaccinated against COVID-19 is that the vaccine is imperfect and allows transmission and infection; at least for these agents, both the rationale and the outcome run counter to that apparently predicted by standard Game Theory approaches?

Line 530. The words “alone” and “vector” need to be swapped in order.

Lines 532-534. This is a nice result. A primia facie argument that CSS would not be worth buying if disease can spread horizontally and spoil one’s investment is enticing, but the results indicate that the conditions for adoption of CSS are more subtle than that; if disease can’t spread it’s not needed, and if it spreads too readily one can’t protect one’s investment. It seems that at least in some of the modeled scenarios the threshold disease spread at which CSS becomes worthwhile should be derivable? Would it be where the expected loss caused by the (uncertain) horizontal spread equals the cost of CSS?

Lines 535-539. I agree with the authors here. The “grower vs alternative” evaluation most closely approximates the approach taken by growers with whom I have worked and observed.

Reviewer #2: The manuscript is overall well written and summarizes a large amount of high-quality work. My comments at this time are fairly general and higher level and meant to improve the overall manuscript.

First, the authors should structure the manuscript as described in the Submission Guidelines. Specifically, the Materials and Methods sections should be moved to the end of the main text. this organization change from the “classic” organization is useful for mathematical model papers of this sort, as it emphasizes the results and discussion so that readers understand the paper’s contributions and can then bury themselves in the details of the modeling if desired. Besides this organizational issue, in my opinion, the paper is trying to do too much. The current version is quite long and yet still too terse in many key parts to understand the findings, their nuances, and their wider connections to the literature.

The overall paper objective (lines 89-91) is good, but achieving it by answering the five questions (lines 91-98) are too many. The main question to examine is “How does including grower behaviour in plant epidemic models affect the outcome of control schemes, particularly when control strategies provide incomplete control?” (lines 89-93). “With incomplete control, under what conditions, if any, can high levels of grower adoption of control strategies be attained and maintained?” “Discussion of results will focus on the implications for encouraging adoption of control strategies. Sensitivity analysis will examine how results respond to changes in key parameters and assumptions.” I recommend dropping some of them or burying more of them in an appendix/supplement (or a second paper) and simply summarize the results of question 1 to 5 in the main text in order to focus on the main point. Right now your main point is buried under a huge pile of equations, variables, tables, figures and text, making it hard to find. Your goal should not be to summarizer all the work you have done (which is a lot), but to communicate your key finding by doing a better job of explaining it, its nuances, connecting it the existing literature and developing policy recommendations.

Second, I recommend tying the human behavior modeling more directly to the existing literature, including explaining and justifying assumptions. For example, equation 33 is simply posited with little explanation or tying to existing papers. See how Milne et al. (2015) (your citation 6) and Saikai et al. 2020 (DOI 10.1002/ps.6016) state and explain/justify it. Once you stop trying to put everything you have done into one paper and instead focus on the main point, you can bring readers along and help them understand what you are doing and connect to the existing academic conversation. There are many examples I could have noted.

Along these same lines, I recommend using what you call “strategy vs” or “grower vs” assumptions, not both, and then connecting them more clearly to terminology used in existing literature. The difference between the two is whether the farmer the previous year’s outcomes to form expected changes in profit or forms expectations using complete knowledge of the system. I think the difference it akin to what macro-economists call adaptive expectations versus rationale expectations. I spent some time looking in the results sections, but I was not able to easily figure out how the results differed for the two assumptions (adaptive vs rationale expectations). I suggest choosing one or the other, not both. Which one has the most empirical support or theoretical support? Which one have most researchers used? Only keep them if you get very different results and have a good reason that both may be accurate assumptions.

In addition, the send part of the comparison (alternative or postulation) I think should be tied to discussions of social networks or how farmers develop their information to use when making decisions. Wah you term “alternative’ is basic expected profit maximization, with no information about how neighbors have done, while “population” assumes the farmer has clear information on what the population average is. Kaup (Rural Sociology 73(1), 2008, pp. 62–81) is a good place to start, Sherman and Gent (2014) http://dx.doi.org/10.1094/PDIS-03-14-0313-FE overview the issues in Plant Disease, and also see Saikai et al. (2020) (DOI 10.1002/ps.6016). In some sense, the choice between adaptive vs rationale expectations is also a similar information problem that Kaup examined. How does a farmer “know’ the real probability of infection and loss? Do they have to rely on their own experience or can they some how get access to the “true” probabilities? The same applies for the comparison group: how do they know the expected profit for the alternative if they have never experienced it or the average for the group as a whole? My recommendation it to connect your assumptions to the existing literature and the terminology that they use. Right now, your work is in some sense in the same vein as these and other papers, but no one would know this via a google or key terms search, as you do not cite the key papers or use their terminology.

Lastly, I am not a fan of the “catch phrase” used in the title. I am not against the catch phrase, but I had to spend some time trying to figure out what exactly it was saying. It reminded me of an Escher sentence or similar. I also am not a fan of the use of the colloquial “ain’t”. I have no recommendation – if you want to use that title, do so, but I thought it best to at least give you my honest feedback about it.

I have not gone into any of the more detailed comments, but leave such comments for later versions once the paper is revised. Overall, I believe that there is a publishable paper buried in this work, but the authors have to stop trying to put everything they have done into one paper, focus on communicating their main point, and dropping material that does not contribute to making this point.

Reviewer #3: Dear authors, please find attached two documents of my

review. A word document with general comments and an

annotated PDF version of the submitted manuscript with

detailed comments.

**Have the authors made all data and (if applicable) computational code underlying the findings in their manuscript fully available?**

Reviewer #1: Yes

Reviewer #2: None

Reviewer #3: **No: **Authors said they will submit the code of their models upon publication

PLOS authors have the option to publish the peer review history of their article (what does this mean?). If published, this will include your full peer review and any attached files.

Reviewer #1: **Yes: **Neil McRoberts

Reviewer #2: No

Reviewer #3: No
---

## [Decision Letter · Decision Letter 1]

26 May 2022

Dear Ms Murray-Watson,

Thank you very much for submitting your manuscript "How grower decision-making is modelled impacts the effect of plant disease control." for consideration at PLOS Computational Biology. As with all papers reviewed by the journal, your manuscript was reviewed by members of the editorial board and by several independent reviewers. The reviewers appreciated the attention to an important topic. Based on the reviews, we are likely to accept this manuscript for publication, providing that you modify the manuscript according to the review recommendations.

Sincerely,

Claudio José Struchiner, M.D., Sc.D.

Associate Editor

PLOS Computational Biology

Tom Britton

Deputy Editor

PLOS Computational Biology

[LINK]

Reviewer's Responses to Questions

**Comments to the Authors:**

Reviewer #1: I am grateful to the authors for their thoughtful and comprehensive response to the suggestions made on the previous draft of the paper. I have no further comments.

Reviewer #2: First, I have a quick title suggestion: “How growers make decisions impacts plant disease control”. Second, I still think using “strategy vs” and “grower vs” are poor choices to describe the two classes models. Imagine if you were someone for whom English was a second language, you read this article and liked it and then wanted to talk to native speakers in the field about it. If you then started talking to them about a “strategy vs” model or a “grower vs” model, thinking this was a standard way of talking about such models, no one would know what you were talking about. I suggested finding a better terminology for the purpose of helping people understand and communicate about your paper. As it stands, the current terminology is an impediment, even among native speakers.

Micro comments

Line 29: drop “do”

Line 111: use “a different”, not “an”

Line 351: something is missing, maybe “controlling it does not …”?

**Have the authors made all data and (if applicable) computational code underlying the findings in their manuscript fully available?**

Reviewer #1: Yes

Reviewer #2: None

PLOS authors have the option to publish the peer review history of their article (what does this mean?). If published, this will include your full peer review and any attached files.

Reviewer #1: **Yes: **Neil McRoberts

Reviewer #2: No

Figure Files:

Data Requirements:

Reproducibility:

References:

---

## [Editor Report · Decision Letter 2]

16 Jun 2022

Dear Ms Murray-Watson,

We are pleased to inform you that your manuscript 'How growers make decisions impacts plant disease control.' has been provisionally accepted for publication in PLOS Computational Biology.

Best regards,

Claudio José Struchiner, M.D., Sc.D.

Associate Editor

PLOS Computational Biology

Tom Britton

Deputy Editor

PLOS Computational Biology

---

## [Editor Report · Acceptance letter]

2 Aug 2022

PCOMPBIOL-D-21-02295R2 

How growers make decisions impacts plant disease control.

Dear Dr Murray-Watson,

I am pleased to inform you that your manuscript has been formally accepted for publication in PLOS Computational Biology. Your manuscript is now with our production department and you will be notified of the publication date in due course.

With kind regards,

Zsofia Freund
